# Circulating Plasma Syncytin-1 mRNA in Preeclampsia—A Pilot Study

**DOI:** 10.3390/ijms26189015

**Published:** 2025-09-16

**Authors:** Lisa Lorenz-Meyer, Christoph Bührer, Stefan Verlohren, Stefanie Endesfelder

**Affiliations:** 1Department of Obstetrics, Charité—Universitätsmedizin Berlin, 13353 Berlin, Germany; lisa.lorenz-meyer@charite.de; 2Department of Neonatology, Charité—Universitätsmedizin Berlin, 13353 Berlin, Germany; christoph.buehrer@charite.de; 3Department of Obstetrics and Fetal Medicine, University Medical Center Hamburg-Eppendorf, 20251 Hamburg, Germany; s.verlohren@uke.de

**Keywords:** preeclampsia, syncytin-1, biomarkers, placental dysfunction, cell-free fetal RNA

## Abstract

The endogenous human retroviral protein syncytin-1 is vital for placental integrity via syncytiotrophoblast formation. Preeclampsia has been associated with reduced placental syncytin-1 expression. The goal of the prospective pilot study was to investigate the role of circulating fetal syncytin-1 mRNA in women with and without preeclampsia (PE). We therefore determined syncytin-1 mRNA in maternal plasma of n = 26 women without and n = 43 women with PE with quantitative real-time PCR. Median circulating syncytin-1 mRNA concentrations were significantly lower in women with PE [1 217 050 copies; IQR 610 900–1 482 069] vs. those without PE [1 992 739 copies; IQR 1 616 811–3 126 225], *p* < 0.001. The area under the receiver operating curve used to diagnose PE at the time of delivery was 0.913 (95% CI: 0.850–0.977). Sub-analysis of patients with PE at sampling and patients who later developed PE vs. gestational-age-matched controls yielded a diagnostic accuracy of fetal syncytin-1 expression with an AUC of 0.962 (95% CI 0.877–1.000) and a predictive accuracy of 0.924 (95% CI 0.842–0.983). Quantification of the fetal syncytin-1 expression in maternal plasma might be used as a diagnostic and predictive tool in PE.

## 1. Introduction

With an incidence of 2–8%, preeclampsia (PE) contributes to significant maternal and neonatal morbidity and mortality worldwide [1]. Optimal management of patients at risk therefore requires early diagnosis and prediction of the disease [2].

For this purpose, the accuracy of several diagnostic and predictive biomarkers has been investigated in recent decades [3,4]. Potential sources of biomarkers indicating PE and associated pregnancy complications rely on the pathogenesis of the disease, which occurs due to a placental insufficiency with generalized endothelial inflammation and dysfunction affecting various organs [5]. MacDonald et al. summarized the most promising markers and highlighted the usefulness of the placental proteins soluble fms-like-tyrosinekinase (sFlt-1) and Placental Growth Factor (PlGF) [4]. The need for further reliable biomarkers in diagnosing and predicting PE led to the aim of the present study, which investigates the use of fetal syncytin-1 mRNA in maternal plasma as a biomarker in PE.

Syncytin-1, an envelope protein encoded by the endogenous retrovirus ERVW-1, plays a critical role in human placental development by mediating the fusion between human trophoblasts and endometrial stromal cells [6] and formation of the syncytiotrophoblast layer [6]. This structure is essential for maternal–fetal exchange. Syncytin, a protein of fetal origin, is present in maternal circulation exclusively within syncytiotrophoblast-derived extracellular vesicles, with no evidence of free plasma presence or maternal tissue expression [7,8]. Dysregulation of syncytin-1 expression or function has been linked to hypertensive pregnancy disorders such as PE [9,10].

In PE, impaired syncytin-1 activity often coincides with sterile inflammation and elevated expression of antiviral proteins [9,10]. These restriction factors, known to inhibit viral envelope function, may disrupt syncytin-1-mediated fusion by interfering with Env protein processing, maturation, or membrane incorporation, suggesting a mechanistic link between inflammation and placental dysfunction.

As shown by single-cell transcriptomic analyses, a widespread dysregulation of syncytiotrophoblast and extravillous trophoblast gene expression, including downregulation of syncytin-1, whether by epigenetic silencing, genetic downregulation, or interference from inflammation-induced restriction factors, impairs trophoblast fusion and syncytiotrophoblast integrity [11,12,13]. As a consequence of the reduced syncytin-1 expression and impaired cell fusion, a disorganized and thinned syncytiotrophoblast layer, reduced placental barrier and exchange capacity with abnormal placental vascularization can be observed as a causal placental pathology in PE and FGR [14,15,16].

Previous investigations indicate that in PE, both, the concentration and molecular profile of syncytin-1-positive microvesicles are markedly altered, which is why several authors suggest measuring the fetal syncytin-1 level in the maternal plasma as a possible accurate biomarker for the disorder [8,17].

The objective of this study was to comprehensively evaluate the expression levels of fetal syncytin-1 in plasma from pregnant women with and without PE, and to assess the predictive and diagnostic potential of syncytin-1 for PE.

## 2. Results

### 2.1. Study Population

Between 19 June 2019 and 14 May 2022, a total of 87 patients were included in the study, and n = 18 were lost to follow-up. Of the 69 study participants, n = 26 had a final diagnosis of PE, n = 17 already had the diagnosis at time of blood sampling, and n = 9 had only symptoms of PE at time of sampling and later developed PE. A total of n = 43 had no PE. Baseline characteristics, PE-related symptoms at time of inclusion of the women with and without PE, and pregnancy outcome are presented in Table 1. Patients with PE presented at a median gestational age of 32 + 5 weeks (IQR: 26 + 6–35 + 2), and patients with no PE at 28 + 5 weeks (IQR: 25 + 1–35 + 3, *p* = 0.093). According to the delivery outcome in both groups, PE-related symptoms such as hypertension (media systolic blood pressure in PE vs. no PE 150.0 mmHg (IQR: 142.0–160.0 vs. 116.0 mmHg (IQR: 110.0–127.0), *p* < 0.001; diastolic blood pressure (96.0 mmHg, IQR 84.0–100.0, vs. 68.0, IQR: 59.0–73.0, *p* < 0.001)), proteinuria (positive dipstick readings (++ or higher) in 50% of PE cases, none in no-PE cases (*p* < 0.001)), a higher pulsatility index of the uterine arteries (1.30, IQR 1.11–1.69 in PE group vs. 0.84, IQR 0.70–1.38 in no-PE group, *p* = 0.01), and suspected FGR (34.6 vs. 11.6%, *p* = 0.031) were more frequent in the PE group.

### 2.2. Fetal Syncytin-1 mRNA and sFlt-1/PIGF

The study included 69 participants: 43 without preeclampsia (no PE) and 26 diagnosed with PE (Table 2). The sFlt-1/PlGF ratio was available in n = 36 women without PE and in n = 24 patients with PE. Serum levels of sFlt-1, PlGF, and the sFlt-1/PlGF ratio differed significantly between groups. In no-PE women, median sFlt-1 was 2 295.5 pg/mL (IQR: 1 605.8–3 661.8), PlGF was 232 pg/mL (IQR: 116.5–348.5), and the sFlt-1/PlGF ratio was 10.5 (IQR: 5.25–27.25). In contrast, PE patients showed elevated sFlt-1 (8 289.5 pg/mL; IQR: 6 418.0–13 496.8), reduced PlGF (43.7 pg/mL; IQR: 24.0–78.3), and a markedly increased sFlt-1/PlGF ratio (214.0; IQR: 142.0–371.5), all with *p <* 0.001.

As shown in Figure 1A, the sFlt-1/PlGF ratio was substantially higher in PE cases, with greater variability, indicating significant antiangiogenic imbalance. The no-PE group had lower, stable ratios, consistent with physiologic angiogenesis. Similarly, fetal syncytin-1 mRNA copy numbers were significantly lower in the PE group (median 1 217 050; IQR: 610 900–1 482 069) compared to the no-PE group (median 1 992 739; IQR: 1 616 811–3 126 225; *p* < 0.001). As shown in Figure 1B, syncytin-1 expression was reduced and showed greater variability in PE, indicating impaired and heterogeneous placental function.

### 2.3. Syncytin-1 mRNA Expression over Time Prior to Preeclampsia Diagnosis

To highlight differences in syncytin-1 mRNA copy numbers between women without PE (no-PE) and those later diagnosed with PE, we analyzed syncytin-1 copy numbers across four gestational age ranges (<28, <32, <36, and ≤40 weeks). In all gestational categories, mean syncytin-1 levels were consistently lower in PE compared to no-PE groups (Figure 2). Notably, in the <28-week, 32–36-week, and 36–40-week groups, syncytin-1 mRNA copy number was significantly reduced in PE patients in cases of symptom-based suspicion of preeclampsia compared to no-PE women. Expression in no-PE samples remained stable, while PE samples displayed greater variability. As already described above, of n = 26 patients in the PE group, n = 9 patients only developed PE within a median time of 17 days (IQR: 3.5–63.5) after sampling of the analytes. Figure 3 displays the syncytin-1 expression levels in correlation with the remaining time to diagnosis of PE, which indicates that syncytin-1 expression decreases with the oncoming onset of the disease.

### 2.4. Reduced Maternal Syncytin-1 Expression Before and After Onset of PE and Its Diagnostic and Predictive Potential

To determine the performance of fetal syncytin-1 and the sFlt-1/PlGF ratio to distinguish women with and without PE, we generated corresponding receiver operating characteristic (ROC) curves with area under the curve (AUC) in the whole cohort. As shown in Figure 4, the AUC of fetal syncytin-1 was 0.913 (95% CI: 0.850–0.977; n = 43 no PE, n = 26 PE), while the AUC for sFlt-1/PlGF was 0.970 (95% CI: 0.930–1.000; n = 36 no PE, n = 24 PE). Both biomarkers therefore demonstrated excellent discriminative power (*p* < 0.0001 for both comparisons).

Further sub-analysis of the patients with PE at sampling and a gestational-age-matched cohort with no PE (n = 31), as well as of patients with PE later in pregnancy and a gestational-age-matched cohort with no PE (n = 26), underlines the diagnostic and predictive potential of fetal syncytin-1. As shown in Table 2, patients with onset of PE at or before sampling and those with a later diagnosis of PE had a significantly decreased fetal syncytin-1 expression when compared to the no-PE group (PE diagnosed at sampling: median 1 249 321 copies, IQR 616 665–1 477 892 vs. no PE: median 2 025 940 copies; IQR 1 629 823–2 396 807, *p* < 0.001, PE diagnosed at a later gestational age: median 954 986 copies; IQR 427 006–1 434 322 vs. no PE: median 2 031 055 copies; IQR 1 888 707–281 675, *p* < 0.001). The respective AUCs were 0.962 (95% CI 0.877–1.000) for the diagnosis of PE with the fetal syncytin-1 copy number and 0.924 (95% CI 0.842–0.983) to predict PE.

In the last step, we tested Spearman’s correlation between fetal syncytin-1 as a new biomarker for PE and the sFlt1–1 and PlGF concentrations, as well as the sFlt-1/PlGF ratio. A significant correlation between PlGF and syncytin-1 copy number (ρ = 0.552, 95% CI 0.341–0.711) could be shown, as well as an inverse correlation between sFlt-1 and fetal syncytin-1 copy number (ρ−0.491, (95% CI −0.667–−0.264), and the sFlt-1/PlGF ratio (ρ = −0.574, 95% CI −0.726–−0.368).

## 3. Discussion

This investigation showed that the circulating fetal syncytin-1 mRNA level in maternal plasma is not only significantly reduced in women with PE at or after diagnosis, but also before onset of the disease, when compared with no PE. This goes along with an inverse linear correlation with the established predictive and diagnostic biomarker sFlt-1/PlGF.

As depicted across different gestational ages in our study in women with no PE, circulating fetal syncytin-1 messenger RNA remains robust throughout pregnancy. The higher elevated syncytin-1 expression in pregnancies without PE may reflect normal placental development and transcriptional activity [11].

The finding of markedly decreased maternal plasma fetal syncytin-1, even in women ≤28 weeks who develop PE, not only indicates the potential clinical significance as an additional predictive biomarker but is also consistent with the established model that early gestational molecular disturbances leading to impaired trophoblast fusion and syncytiotrophoblast dysfunction are central to the pathogenesis of PE [18,19,20]. The observed variability in syncytin-1 levels within the PE group, especially in late pregnancy, supports the concept that PE is a heterogeneous disorder comprising multiple subtypes with distinct molecular profiles [11,21,22,23].

Our results on maternal plasma syncytin-1 confirm previous reports on placental syncytin-1 association with PE: Multiple studies demonstrated that syncytin-1 mRNA and protein are decreased in preeclamptic placentas, suggesting a downregulation of syncytin-1 expression in the context of PE with compromised placental development or function. This correlates with impaired trophoblast fusion and syncytiotrophoblast formation, which are key processes for a normal placental barrier and exchange function [11,24,25].

As in our study, recent large-scale plasma cell-free RNA studies have moreover shown that fetal syncytin-1 mRNA is already decreased in maternal circulation before PE becomes clinically apparent [19]. This also supports its potential as an early predictive biomarker, like in our pilot study with a receiver operating characteristic (ROC) area under the curve (AUC) exceeding 0.9, comparable to the clinically established sFlt-1/PlGF ratio [19].

To date, measurement of the PlGF level is recommended in the first trimester to identify women at risk for PE who could benefit from early aspirin prophylaxis, whereas the sFlt-1/PlGF ratio has the greatest negative predictive value (NPV) up to 4 weeks before the onset of PE after 20 + 0 weeks [3,26]. A second independent biomarker, especially in women with unclear symptoms in the second and early third trimester, might be able to improve the positive predictive value (PPV) and the predictability of the time until onset of the disease. Similarly to the sFlt-1/PlGF ratio, previous studies have shown a correlation between syncytin-1 mRNA levels and PE severity, suggesting that this biomarker could aid in clinical decision-making including inpatient admission. However, these previous findings stemmed from small case–control studies or experimental models. Therefore, our pilot study further verifies the usefulness of this PE biomarker, although future large, multicenter clinical cohorts are needed to establish and validate its routine clinical utility. [7,8,27].

Our study has limitations. As a pilot study, the investigation included a relatively small cohort size and high percentage of cases lost to follow-up that were controls presenting for routine examinations. Of the present PE cohort, 65.4% were early-onset cases ≤34 weeks of pregnancy; differences in the biomarker levels between controls and only women with late-onset PE might be less significant. A validation of our findings in larger, multicenter studies to confirm the robustness and the predictive value of syncytin-1 mRNA as a biomarker would therefore be desirable.

Possible confounding conditions that may affect the fetal syncytin-1 expression level are the gestational age, systemic inflammation or immune activation, and epigenetic modifications. The increasing syncytin-1 expression from the first trimester until term that other authors describe would suggest the use of Multiple of the Median (MoM) values to avoid resulting inaccuracies [28].

In regard to the pregnancy outcomes of our study, only 49% of the no-PE group delivered their babies at term after 37 + 0 weeks, of these 49.1% due to preterm contractions and/or rupture of the membranes. This means that acute or chronic inflammation with cytokine-mediated altered syncytin-1 expression might have influenced these cases [15,29]. Other preconditions such as metabolic syndrome with obesity and diabetes mellitus are also linked to chronic low-grade inflammation, which might have impaired trophoblast function and syncytin-1 mRNA levels [15].

Moreover, a hypoxic state of the placenta with subsequent suppression of syncytin-1 expression occurs not only in PE but also in PE-associated complications including FGR [11,28]. Since there were also FGR cases in the PE and no-PE groups, further differentiation between women with and without FGR would be desirable in future studies. However, only one patient without PE required an FGR-related induction of labor, whereas all patients of the PE group had a PE-related iatrogenic termination of pregnancy (n = 2 of these also because of an FGR). This underlines even more that the fetal syncytin-1 expression indicates a placental insufficiency including maternal PE and fetal FGR, which might be useful to predict imminent PE-related delivery [25].

## 4. Materials and Methods

### 4.1. Study Population

All patients were enrolled at the in- and outpatient Perinatal Center of the Charité University Medicine Berlin. Women with signs and symptoms of PE, either with newly diagnosed or suspected PE, presenting to the inpatient clinic were enrolled. As gestational-age-matched controls, we included hospitalized pregnant women without PE (no PE). Furthermore, healthy pregnant women attending routine prenatal screenings during the second or third trimester were included to establish reference values for maternal serum syncytin-1 concentrations.

Inclusion criteria comprised pregnant women aged between 18 and 45 years with singleton pregnancies who had provided written informed consent. Patients with newly diagnosed or suspected PE were included in the study, if they showed signs and symptoms of PE such as new onset of hypertension, worsening of pre-existing hypertension, and/or if they showed one or more of the following clinical signs suggestive of PE: new-onset proteinuria, persistent headache, upper abdominal pain, severe edema, excessive weight gain (≥2 kg per week), or a pathological mean pulsatility index (PI) of the uterine arteries (defined as mean PI > 1.6).

Exclusion criteria included fetal malformations, fetal aneuploidies, maternal autoimmune diseases, acute fatty liver of pregnancy, hemolytic–uremic syndrome, and loss to follow-up. The study was approved by the institutional ethics committee (EA2/246/18).

### 4.2. Study Design and Definition of Outcome Groups

This prospective pilot study was conducted to evaluate maternal blood syncytin-1 concentrations in pregnant women with no PE compared to those with either a subsequent development or a new diagnosis of PE. Due to the exploratory nature of the study, no predefined protocol was implemented, and the timing of blood sampling was not standardized. All participants received routine clinical care in accordance with established institutional and national guidelines. PE was defined according to the 2012 definition of the International Society of the Study of Hypertension in Pregnancy and the 2003 definition of the American College of Obstetricians and Gynecologists as hypertension ≥140/90 mmHg in combination with proteinuria of ≥300 mg/d or 2+ positive urine dipsticks after 20 + 0 weeks of pregnancy [30,31]. Fetal growth restriction (FGR) was defined as an estimated fetal weight below the 10th percentile and/or evidence of growth deviation from the individual growth trajectory during pregnancy, in combination with at least one of the following findings: abnormal Doppler indices of the umbilical or uterine arteries, or the presence of oligohydramnios [32]. Fetal weight percentiles were calculated using the integrated reference values provided by the ultrasound software Viewpoint 5 (GE Healthcare, Düsseldorf, Germany) [33].

### 4.3. Isolation of Fetal RNA from Maternal Whole Blood Samples

Fetal syncytin-1 mRNA was quantified from maternal whole blood collected via venipuncture in EDTA-coated tubes (Sarstedt, Germany) and stored at −80 °C until analysis. RNA isolation was performed using the GenUP Virus RNA Kit (biotechrabbit GmbH, Berlin, Germany) according to the manufacturer’s protocol with minor modifications. Briefly, 300 µL whole blood was processed per sample, followed by lysis, binding, washing, and elution steps using spin columns. RNA was eluted in 50 µL preheated buffer and stored at −80 °C. Quantification was conducted postnatally and had no influence on clinical management. A detailed protocol is provided (see Appendix A).

### 4.4. Reverse Transcription and Pre-Amplification

Reverse transcription of 200 ng total RNA was performed using the FastGene^®^ Scriptase II cDNA Synthesis Kit (Nippon Genetics, Düren, Germany) following the manufacturer’s instructions. The resulting cDNA was stored at −20 °C. For increased sensitivity, cDNA was pre-amplified with the TaqMan PreAmp Master Mix Kit (Applied Biosystems, Waltham, MA, USA) using custom syncytin-1 primers. Pre-amplification was carried out over 10 PCR cycles and the products were diluted 1:5 before quantitative PCR. A detailed protocol is provided (see Appendix A).

### 4.5. Quantitative PCR with Calibration Curve for Absolute Copy Number Determination

Quantitative real-time PCR (qPCR) was performed using the TaqMan Fast Advanced Master Mix (Applied Biosystems) and a custom TaqMan Gene Expression Assay targeting syncytin-1. Reactions (20 µL) were run in triplicates on 96-well plates, using 1 µL pre-amplified cDNA per reaction. Absolute quantification was based on a standard curve generated from serial dilutions (10^3^–10^11^ copies) of synthetic customized SYN1 dsDNA (IDT, Integrated DNA Technologies, Coralville, IA, USA). Further details are provided (see Appendix A).

### 4.6. Quantification of the sFlt-1/PlGF Ratio in Maternal Serum

sFlt-1 and PlGF concentrations were determined on the Cobas platform (Roche Diagnostics GmbH, Mannheim, Germany) in our affiliated laboratory. Since this is a biomarker of the clinical routine, patients, attending physicians, nurses, and midwives were aware of the sFlt-1/PlGF ratio [3].

### 4.7. Doppler Ultrasonographic Evaluation

Doppler ultrasonography was performed in accordance with the guidelines of the Fetal Medicine Foundation, including assessment of the mean pulsatility index (PI) of the uterine arteries, umbilical artery, and fetal middle cerebral artery. Examinations were conducted at the time of sFlt-1/PlGF ratio measurement, with a tolerance of ±2 days [34].

### 4.8. Statistical Analysis

The patient’s data were collected from medical records in our data system Viewpoint by GE (Solingen, Germany) or SAP (Walldorf, Germany). Analyses were performed with R (R Core Team, Version (2021). R: A language and environment for statistical computing. R Foundation for Statistical Computing, Vienna, Austria. URL https://www.R-project.org/ (accessed on January 8, 2025), just as additional R packages [35,36,37]) and with the IBM Statistical Package for Social Sciences, Version 28.0 (SPSS, IBM Corp., Armonk, NY, USA). Median and interquartile ranges (IQRs) were used as indicated for continuous variables, and numbers and frequencies (%) for categorical variables. Missing data are indicated in the tables. Test selection was based on the evaluation of the variables for normal distribution. The Mann–Whitney U-test was used as appropriate for continuous variables, while two-tailed Fisher’s exact test was used to analyze categorical variables. Subgroup analysis of syncytin-1 expression across gestational age was performed using the Kruskal–Wallis test, followed by multiple mean comparisons using the Bonferroni post hoc test. *p*-values for all tests were two-sided, and statistical significance was set at α ≤ 0.05. ROC curves were used to evaluate the diagnostic performance of syncytin-1 and the sFlt-1/PlGF ratio for PE, with AUCs and 95% CIs reported. Spearman correlations between the sFlt-1/PlGF ratio and fetal syncytin-1 mRNA were calculated, including 95% confidence intervals. Visualizations were performed using GraphPad Prism 8.0 software (GraphPad Software, La Jolla, CA, USA).

## 5. Conclusions

In summary, we can conclude that fetal syncytin-1 mRNA is a potential biomarker to diagnose and predict women with PE and PE-related pregnancy outcomes. Its added value, especially early in pregnancy where established markers such as the sFlt-1/PlGF ratio do not show a sufficient negative and positive predictive value and accuracy, should be evaluated in further prospective studies.

## Figures and Tables

**Figure 1 ijms-26-09015-f001:**
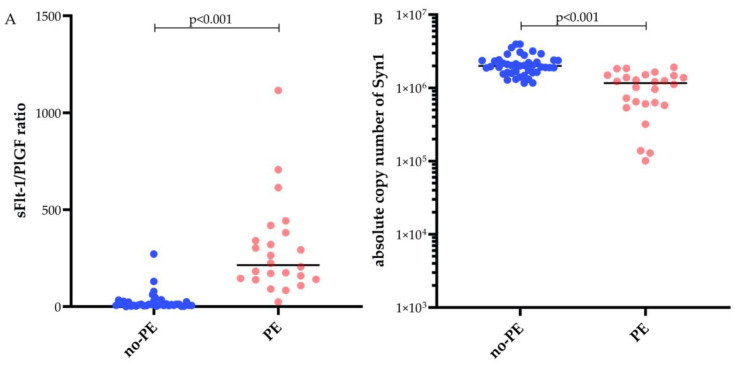
Scatter plots comparing the sFlt-1/PlGF ratio and absolute copy number of the biomarker syncytin-1 (Syn1) in maternal plasma from pregnancies with and without preeclampsia. (**A**) sFlt-1/PlGF ratio measured in maternal plasma samples from pregnancies without preeclampsia (no-PE, n = 34) and with preeclampsia (PE, n = 23), and (**B**) absolute copy number of syncytin-1 (Syn1) measured in plasma from no-PE (n = 43) and PE (n = 26) pregnancies.

**Figure 2 ijms-26-09015-f002:**
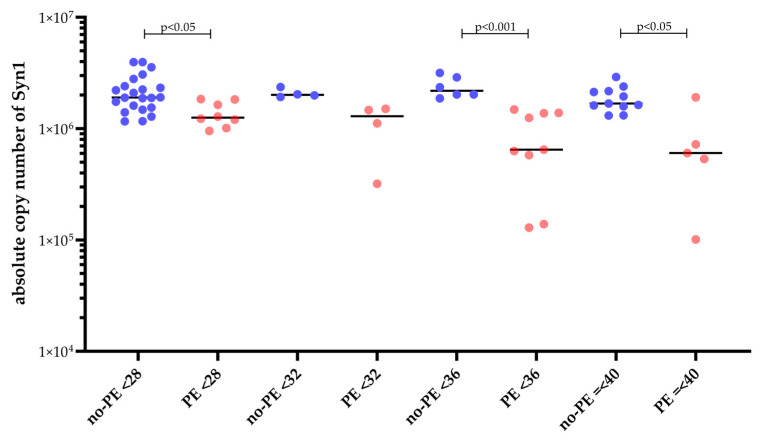
Syncytin-1 mRNA copy numbers in normotensive (no PE, n = 43) and preeclamptic (PE, n = 26) pregnancies across five gestational age groups (<28, <32, <36, and ≤40 weeks). PE samples showed consistently lower expression than no-PE controls, with significant reductions at <28 (*p* < 0.05), <36 (*p* < 0.001), and <40 (*p* < 0.05) weeks, as determined by the Kruskal–Wallis test followed by Bonferroni post hoc testing. Expression in no-PE samples remained stable, while PE samples displayed greater variability.

**Figure 3 ijms-26-09015-f003:**
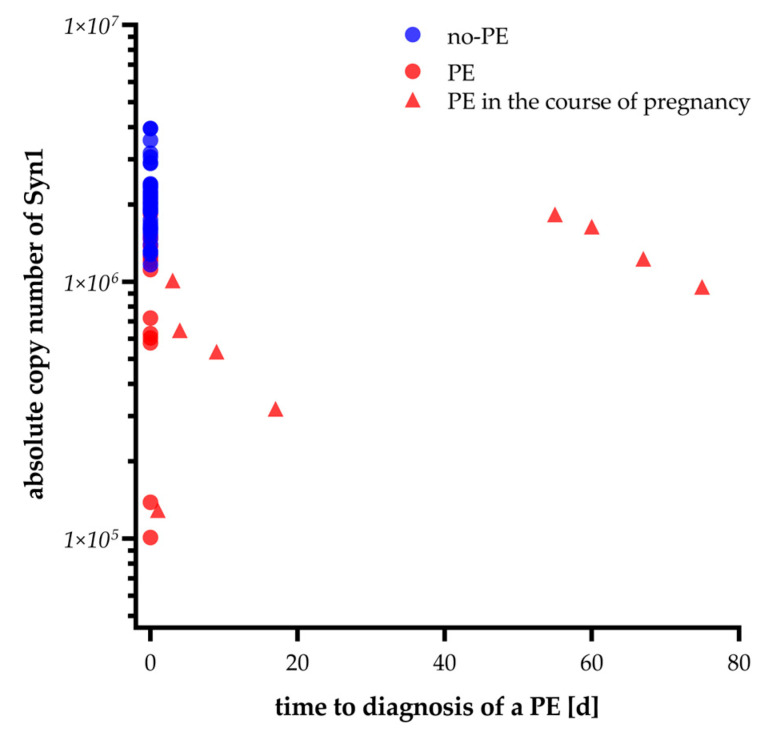
Maternal syncytin-1 mRNA concentrations in pregnancies later complicated by preeclampsia in no-PE (n = 43) and PE patients (n = 26), plotted against the remaining time to diagnosis. The data suggest a decreasing trend in syncytin-1 mRNA levels with shorter intervals to preeclampsia diagnosis.

**Figure 4 ijms-26-09015-f004:**
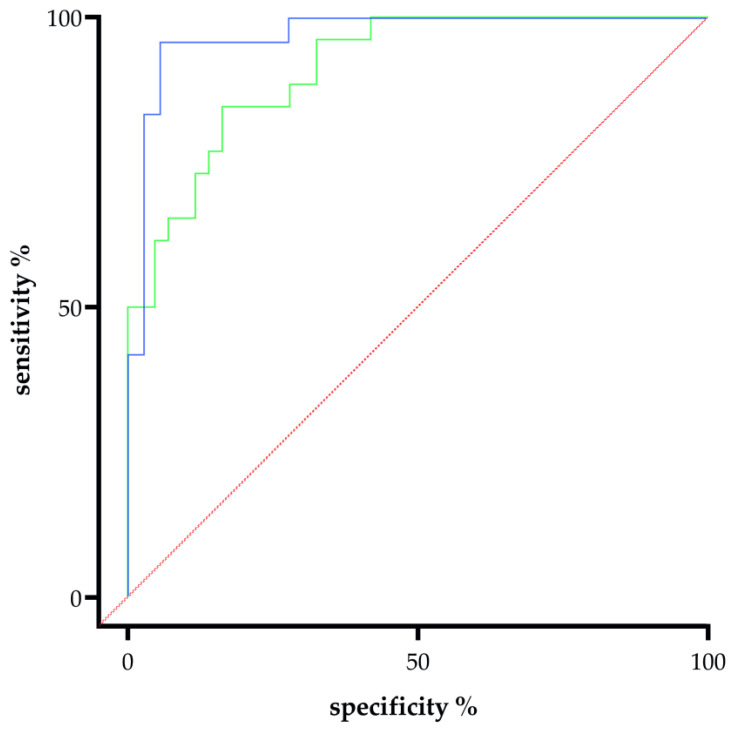
Receiver operating characteristic (ROC) curves comparing the diagnostic performance of syncytin-1 (Syn1, green) and the sFlt-1/PlGF (blue) ratio for distinguishing preeclampsia (PE) from normotensive pregnancies (no PE). The area under the curve (AUC) for Syn1 was 0.913 (95% CI: 0.850–0.977; n = 43 no PE, n = 26 PE), while the AUC for sFlt-1/PlGF was 0.970 (95% CI: 0.930–1.000; n = 36 no PE, n = 24 PE). Both biomarkers demonstrated excellent discriminative power (*p* < 0.0001 for both comparisons).

**Table 1 ijms-26-09015-t001:** Characterization of the total study population.

	No PE n = 43	PE n = 26	*p*-Value
Maternal characteristics			
maternal age, years	33.0 (28.0–26.0)	35.0 (29.5–36.0)	0.471
body mass index, kg/m^2^	26.3 (23.0–30.8)	27.0 (23.3–33.7)	0.665
nulliparity	23 (53.5%)	19 (73.1%)	0.131
ethnicity			
Caucasian	40 (93%)	23 (88.5%)	0.665
black	3 (7%)	1 (3.8%)	>0.999
unknown	-	2 (7.7%)	0.139
PE in previous pregnancy	-	4 (15.4%)	0.017 *
antiphospholipid syndrome	-	-	>0.999
pre-existing kidney disease	-	-	>0.999
maternal age > 40 years	2 (4.7%)	3 (11.5%)	0.358
familiar risk	2 (4.7%)	-	0.523
pre-existing diabetes	-	-	<0.999
BMI > 30	13 (30.2%)	5 (19.2%)	0.402
conception mode			
ICSI or IVF	4 (9.3%)	3 (11.5%)	<0.999
gestational age at sampling, weeks + days	28 + 5 (25 + 1–35 + 3)	32 + 5 (26 + 6–35 + 2)	0.093
patients presenting −20 weeks	7 (16.3%)	-	0.040
patients presenting 20–24 + 0 weeks	5 (11.6%)	3 (11.5%)	>0.999
patients presenting 24 + 1–34 + 0 weeks	20(46.5%)	14 (53.8%)	0.624
patients presenting 34 + 1–37 + 0 weeks	2 (4.7%)	6 (23.1%)	0.046
patients presenting > 37 + 1 weeks	9 (20.9%)	3 (11.5%)	0.514
**Maternal symptoms at sampling**			
systolic blood pressure, mmHg	116.0 (110.0–127.0)	150.0 (142.0–160.0)	<0.001 *
	(n = 16 missing)	(n = 1 missing)	
diastolic blood pressure, mmHg	69.0 (59.0–73.0)	96.0 (84.0–100.0)	<0.001 *
	(n = 16 missing)	(n = 1 missing)	
mean PI of uterine arteries	0.84 (0.70–1.38)	1.30 (1.11–1.69)	0.010 *
	(n = 22 missing)	(n = 8 missing)	
urine dipstick			
negative	35 (81.4%)	4 (15.4%)	<0.001 *
traces	1(2.3%)	2 (7.7%)	0.552
+	-	7 (26.9%)	<0.001 *
++	-	6 (23.1%)	<0.001 *
+++	-	1 (3.8%)	0.379
unknown	7(16.3%)	6 (23.1%)	0.535
new onset of hypertension	3 (7.0%)	15 (57.7%)	<0.001 *
new onset of proteinuria	1 2.3%)	9 (34.6%)	<0.001 *
headache	-	7 (26.9%)	<0.001 *
visual disturbances	-	-	>0.999
progressive edemas	-	1 (3.8%)	0.377
weight gain ≥ 1 kg per week	-	-	>0.999
low platelet count < 150/nL	2 (4.7%)	-	0.523
elevated liver enzymes	4 (9.3%)	1 (3.8%)	0.643
upper abdominal pain	1 (2.3%)	3 (11.5%)	0.147
ascites or pleural effusion	-	-	
PI of the umbilical artery	0.95 (0.78–1.13)	1.0 (0.84–1.36)	0.201
	(n = 6 missing)	(n = 4 missing)	
PI of the MCA	1.74 (1.22–2.13)	1.39 (1.30–1.73)	0.090
	(n = 21 missing)	(n = 9 missing)	
suspected FGR	5 (11.6%)	9 (34.6%)	0.031 *
**Pregnancy outcome**			
gestational age at delivery, weeks + days	36 + 5 (29 + 1–38 + 5)	34 + 1 (32 + 2–36 + 6)	0.396
gestational age at delivery ≥ 37 + 0 weeks	21 (48.8%)	5 (19.2%)	0.003 *
interval from sampling till delivery, days	16.0 (3–57)	8.5 (1.0–18.0)	0.133
interval from sampling till diagnosis, days	-	−1.0 (−5.8–5.3)	
early-onset PE ≤ 34 weeks	-	17 (65.4%)	
no-PE nor FGR	37 (86%)	-	
PE and FGR	-	13 (50%)	
HELLP syndrome	-	3 (11.5%)	
FGR without PE	6 (14%)	-	
fetal birth weight, gram	2560.0 (1160.0–3225)	1807.5 (1368.8–2476.3)	0.119
percentile fetus	40 (19–62)	9.5 (3.8–26.3)	<0.001
delivery mode			
spontaneous	18 (41.9%)	5 (19.2%)	0.068
cesarean	22 (51.2%)	20 (76.9%)	0.573
vaginal-operative	3 (7%)	1(3.8%)	>0.999

Medians and interquartile ranges (IQRs) are shown for continuous variables; the Mann–Whitney U-test has been used for comparison between women with and without PE. For categorical variables, numbers and frequencies (%) are shown; two-tailed Fisher’s exact test was used for comparison between both groups. Statistical significance was set at *p* ≤ 0.05 and is indicated with *. Missing values are indicated.

**Table 2 ijms-26-09015-t002:** Fetal syncytin-1 mRNA and sFlt-1/PlGF.

All Study Objectives			
	**No PE** (n = 43)	**PE** (n = 26)	
gestational age at sampling (weeks + days)	28 + 5 (25 + 1–35 + 3)	32 + 5 (26 + 6–35 + 2)	0.093
sFlt-1 [pg/nL]	2295.50 (1605.75–3661.75)	8289.5 (6418.0–1 3496.8)	<0.001 *
PlGF [pg/nL]	232 (116.50–348.50)	43.7 (24.0–78.3)	<0.001 *
sFlt-1/PlGF	10.5 (5.25–27.25)	214.0 (142.0–371.5)	<0.001 *
	(n = 9 missing)	(n = 3 missing)	
syncytin-1 copies	1 992 739	1 217 050	<0.001 *
	(1 616 811–3 126 225)	(610 900–1 482 069)	
**Patients with PE at time of sampling and GA-matched controls**			
	**No PE** (n = 31)	**PE at or before sampling** (n = 17)	
gestational age at	32 + 2 (26 + 0–37 + 3)	33 + 5 (30 + 6–36 + 1)	0.382
sampling (weeks + days) early-onset PE ≤ 34 weeks	min/max 24 + 4 weeks–40 + 2 weeks	min/max 24 + 4 weeks–40 + 1 weeks	
sFlt-1 [pg/nL]	2 560.0 (1 647.75–3 873.00)	8 215.00 (6 305.50–13 431.50)	<0.001
PlGF [pg/nL]	242.00 (125.50–393.25)	41.30 (23.65–57.40)	<0.001
sFlt-1/PlGF	12.00 (5–28.00)	264.00 (152.0–361.00)	<0.001
syncytin-1 copies	2 025 940	1 249 321	<0.001
	(1 629 823–2 396 807)	(616 665–1 477 892)	
**Patients with PE later in pregnancy and GA-matched controls**			
	**No PE** (n = 26)	**development of PE after sampling** (n = 9)	
gestational age at sampling (weeks + days) time to diagnosis of PE (days) early-onset PE ≤ 34 weeks	26 + 1 (24 + 6–32 + 1) min/max 20 + 5–36 + 2 weeks	27 + 0 (22 + 4–33 + 3) min/max 20 + 4- 36 + 2 weeks 17 (3.5–63.5) 4 (44.4%)	0.810
sFlt-1 [pg/nL]	1882.00 (1514.0–4251.0)	8364.00 (6816.00–1,6107.00)	<0.001
PlGF [pg/nL]	248.00 (121.00–409.50)	81.40 (23.10–94.30)	<0.001
sFlt-1/PlGF	9.00 (2–25.5)	171.00 (91.00–442.00)	<0.001
syncytin-1 copies	2 031 055	954 986	<0.001
	(1 888 707–281 675)	(427 006–1 434 322)	

Medians and interquartile ranges (IQRs) are shown for continuous variables and the minimal and maximal range between gestational age at sampling and diagnosis of PE. The Mann–Whitney U-test has been used for comparison between women with and without PE. For categorical variables, numbers and frequencies (%) are shown; two-tailed Fisher’s exact test was used for comparison between both groups. Statistical significance was set at *p* ≤ 0.05 and is indicated with *. Missing values are indicated.

## Data Availability

The data used to support the findings of this study are available from the corresponding author upon request.

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
