# Peer review of "Circulating Plasma Syncytin-1 mRNA in Preeclampsia—A Pilot Study"

_ijms, 2025, doi:10.3390/ijms26189015_

Round 1

Reviewer 1 Report

Comments and Suggestions for Authors

This manuscript evaluates the possible role of fetal syncytin-1 (in maternal circulation) as a predictive biomarker for preeclampsia.  While there is great interest in syncytin-1 and the formation of the syncytiotrophoblast layer in the development of PE, I do not see how measurements taken either at diagnosis or at suspicion of PE can be at all helpful as a predictive biomarker.  This is a major design flaw and, unfortunately, is not addressable.  Nevertheless, the following comments may improve the manuscript should the authors wish to submit elsewhere.

Overall:

  1. Gestational ages at sample collection should be presented for all three groups. Syncytin-1 levels would be expected to vary by GA and could explain results.
  2. Conclusions that syncytin-1 could be used as a predictive or diagnostic biomarker are not supported by this study.
  3. Unclear how this study design is prospective.  Insufficient information to show prospective follow up occurred. 
  4. STROBE diagram is needed. 
  5. Using PTB pregnancies as controls is not ideal as these are also abnormal and may mask differences. 

Methods:

  1. Samples run x3 so what was CV?

Introduction:

  1. Reference needed for statement in lines 48-49.

Results:

  1. Table 1 -- unclear why some variables have "-" but yet have a p-value assigned to them.
  2. Inappropriate to report p-values of 1.  Should report <0.99
  3. All p-values should have the same number of significant digits
  4. Putting N in the columns suggests no missing data for any of the variables shown.  Is this true?
  5. Unclear why so many variables are categorized (e.g., GA, maternal age, etc.). Preferable to report the average and SD.
  6. Asterisk provided for FGR but nothing ever noted. 
  7. In general, table presentations are poor and difficult to read/follow.  
  8. Line 104 - indicates less variability in PE but results show GREATER variability.
  9. Figure 1 indicates Fisher's Exact tests used but do not see any categorical variables. 
  10. Before combining those with sx and those already diagnosed in case group, comparisons should be made between them to see if they are similar enough to combine.
  11. Line 123 - now indicate greater variability in PE -- conflicts with prior statement.
  12. Figure 2 - one way anova seems inappropriate. Suggest nonparametric Kruskall-Wallis.
  13. Lesser effect at earlier GAs suggests that not a good biomarker for early prediction.  
  14. Unclear how syncytin-1 adds anything to the sFLT1/PlGF ratio.  Need to assess added benefit of syncytin-1 since highly correlated.  May not increase predictability we already have with ratio.
  15. Unclear why only present AUC.  What about SE, SP, PPV, NPV?
  16. Results are unconvincing that syncytin-1 is helpful as a biomarker of prediction or diagnosis.

Discussion:

  1. Line 200 -- if already shown that syncytin1 mrna decreased in maternal circulation before PE, unclear what this study adds.  
  2. Repeatedly says "...a PE" -- do not need the "a"
  3. Line 208 -- high correlation with ratio indicates may not be an independent biomarker.
  4. More LTFU in controls is concerning as this is differential and can bias in unpredictable ways.  More work is needed to assess who these people were and why lost.  
  5. Line 224 -- "A regard..." should be "in regard"

Reviewer 2 Report

Comments and Suggestions for Authors

In this original research work, the authors investigated the usefulness of circulating plasma syncytin-1 mRNA as a biomarker for preeclampsia. Syncytin-1 was studied next to other standard biomarkers like sFLT1 and PLGF. While the data presented is highly relevant, the authors need to explain key concepts and answer some critical questions before approval.

Major comments:

  1. The study population, tables of subject characteristics, and sampling methods require further explanations. Specifically:

1A) how were patients recruited? It seems that some patients were recruited at time of PE diagnosis, but this is not clear, and it needs to be clarified. Potentially, the authors recruited PE patients at different gestational ages and tried to obtain gestational-matched controls. And then some GA-matched controls developed PE? (This would be the group of 9 PE patients). This needs to be clarified in the methods, and also in the results section (first paragraph).

1B) If the authors intentionally recruited early-onset PE patients (PE patients diagnosed with PE before 34 weeks GA), please clearly indicate the rationale for excluding late-onset PE, which represent 90% of PE cases. Is there less value in predicting PE in late-onset PE?

  1. Results: Some key results should have been better explained or are missing. For instance:

2A) The authors didn’t study the effect of FGR on syncitin-1 levels, and it might be expected to observe an association, especially with birth percentiles, or EFW at time of sampling.

2B) Figure 2 is interesting as the decrease of syncitin-1 in PE <28 weeks gestation is much smaller than observed at later GA. What could be the molecular mechanism behind this finding?

2C) Figure 2: it seems that syncitin-1 levels at GA 28 are significantly higher in PE compared to GA 36-40? At least, there seems to be a trend. A 2 way ANOVA will likely show a significant interaction between GA and PE in terms of syncitin-1 and should be included.

2C) Multivariable linear regression needs to be performed to find independent variables that could explain the differences in syncitin-1. For example, GA (as stated in 2B) and FGR could be contributing. Other less likely factors could be maternal BMI.

  1. The authors state that this was a ‘pilot study’ (line 214) with a small number of patients. However, throughout the paper, they mention multiple other studies that have found similarly low levels of syncitin-1 in PE. Therefore, the authors need to clearly state the novelty and significance of this study.

Minor comments:

  1. Please check the grammar.
  2. Table 1: Please divide the subject characteristics as ‘maternal’ and ‘fetal’
  3. Table 1 and throughout the manuscript: ‘gestational age at presentation’ was very confusing. I believe the authors mean, ‘gestational age at sampling’. If so, please change the title of this characteristic. Also, this should go right before the next tab of: gestational age at delivery, for comparison. GA at sampling can be put at the end of ‘maternal characteristics’, and GA at delivery at the beginning of the ‘fetal characteristics.
  4. Table 1: please indicate for each characteristic, whether you have the IQR or the % in parenthesis (for example: maternal age, y (IQR), ethnicity, n (%)).
  5. Table 1: What does ‘state after PE’ mean?
  6. Table 2: Patients with PE at time of presentation should be changed to ‘patients with PE at time of sampling, or sample collection’. Also, there is a mistake under GA: it should be weeks and not days.
  7. Table 2: Patients with PE during the pregnancy (once again, all patients with PE have it during pregnancy), should be changed to: ‘Patients that developed PE after sampling’
  8. Why are some data missing for sFLT1 and PLGF? Table 2 states that 9 no-PE and 3-PE are missing. Did the authors not analyze them on their own, or was this test ordered by the physicians?
  9. Discussion, line 192-193: maternal plasma is not an analyte. Please change to: Our results on maternal plasma syncitin-1 confirm previous reports on placental syncitin-1 association with PE.
  10. Discusion, lines 224-230. This paragraph seems to contradict your previous statements. If inflammation regulates syncitin-1, then the preterm no-PE (51%) should have had lower levels of syncitin-1. The authors need to do a multivariable linear regression or study inflammation before making these statements. Based on their results, I would say that inflammation does not regulate syncitin-1, since early GA at sampling had higher syncitin-1 levels for both PE and no-PE
  11. Figure 1 legend: This figure does not show the interquartile range (please delete this statement). Also: delete the phrase: Fisher’s exact test was used for categorical comparisons. This statement does not belong on this figure.
Comments on the Quality of English Language

Construction of the sentences should be improved. For example: they should write 'diagnosis of preeclampsia (PE)', and not diagnosis of a preeclampsia.

Reviewer 3 Report

Comments and Suggestions for Authors

The manuscript is an interesting paper attempting to touch very important topic – identification of new reliable biomarkers for early prediction of preeclampsia. Although the work appears to be expertly done, a few points should be addressed before considering the manuscript for publication:

  1. Taking into account low number of participants, the statement “a pilot study” should be added to the Title.
  2. Table 2: “gestational age at presentation (days)” and “gestational age at presentation (days) time to diagnosis” are not clear. Please, re-phrase.
  3. The normal plasma/serum PlGF levels are known to be different (to decrease) between 24-26 and 33-39 weeks of pregnancy. Please, specify the GA at which the presented values have been obtained in the Table 2.
  4. Lines 111-112: Please, check “Data are presented as individual values with median and interquartile range”. I don’t see interquartile ranges, the graphs look as scatter plots.
  5. Line 182: “The stable and elevated syncytin-1 expression”: “elevated” is not applicable for normal physiological values.
  6. The text requires a thorough English revision. For example:
  • The third sentence of Abstract appears to lacks the word “goal”.
  • Line 26: might be used.

The same concerns an entire text.

Comments on the Quality of English Language

English improvement is necessary.

Round 2

Reviewer 2 Report

Comments and Suggestions for Authors

The authors answered appropriately the majority of my concerns.

However, a key issue of explaining the novelty and significance of their study is still missing. From what they attempted to explain, I gather that they are saying their study is larger than previously small experimental studies on syncitin-1, but that larger validations studies are required in the future.

I recommend that they change lines 208-212: Likewise, the sFlt-1/PlGF-ratio, previous studies showed a correlation between syncytin-1 mRNA levels and disease severity, which might aid clinical decision-making, including inpatient admission. However, these findings stem from small case-control studies or experimental models, and Syncytin-1 has not yet been validated in large, multicenter clinical cohorts, leaving its routine clinical utility unestablished. [7,8,27]. Please change to:

Likewise to the sFlt-1/PlGF ratio, previous studies have shown a correlation between syncytin-1 mRNA levels and PE severity, suggesting that this biomarker could aid in clinical decision-making including inpatient admission. However, these previous findings stemmed form small case-control studies or experimental models. Therefore, our pilot study further verifies the usefulness of this PE biomarker, although future large, multicenter clinical cohorts are needed to establish and validate its routine clinical utility [7,8,27]. 
